# OpenReview forum: "Shallow Alignment, Deep Deception at Scale: The Evaluation Paradox of Parameter-Efficient Fine-Tuning"
_ICLR.cc/2026/Conference — ICLR 2026 Conference Withdrawn Submission_

### Official Review · Reviewer_78Pd · 2025-10-20

**Soundness:** 1
**Presentation:** 3
**Contribution:** 2
**Rating:** 2
**Confidence:** 5

**Summary:**

The authors investigate the effectiveness of safety alignment across LLMs with different fine-tuning extent and capacity. When stress-tested with ethical reasoning, high-capacity models benefit from deep alignment, whereas shallow alignment tends to induce deceptive behavior. In contrast, for low-capacity models, shallow alignment seems sufficient. Moreover, the superficially aligned model could outperform the robustly aligned model on certain standard safety benchmarks, suggesting a gap between existing benchmarks and real-world use cases, which the partial alignment methods like PEFT may fail to bridge.

**Strengths:**

1. The research focus, i.e., superficial/shallow alignment, is a significant problem as LLMs are increasingly deployed in real-world applications. This paper clearly explains the problem and its importance in the introduction, providing a fresh perspective to inform related solutions.
2. The authors investigate the relationship between model capacity and fine-tuning depth instead of treating them as isolated factors.
3. The proposed practices are grounded in psychology theoretical frameworks.

**Weaknesses:**

1. The text in Fig. 1 should be more legible and better aligned with the caption.
2. My major concern is whether the experiments are sufficient to support the hypothesis.

    i) The experiments are not designed with a single-variable approach. For instance, it would be better to conduct both deep and shallow alignment on the same model to better isolate the effect of alignment depth. However, your current setup includes four varying factors: model family, fine-tuning depth, capacity, and language.

    ii) The classification of the models doesn't make sense, as (a) there are four high-capacity (shallow) models but only one low-capacity (deep) model, which creates an imbalanced setting; (b) I am not confident that the same classification results for model capacity could be reproduced given the differences in their baseline performance.

    iii) Most conclusions are drawn from very few observations. For example, garak-do-not-answer --> standard safety metric, qwen-3-0.6b-chat --> low-capacity model, GPT-4o w/ SFT+RLHF ---> deep alignment. I am unsure about the statistical significance of these conclusions.

**Questions:**

1. The first sentence of the abstract reads a bit awkward to me, since AI alignment encompasses much more than safety. Do you mean that the relationship between fine-tuning extent and model capacity is a significant confounder in the brittleness of AI safety alignment, yet remains underexplored?
2. Could you clarify why these particular models were chosen to represent low/high-capacity categories, and why the number of tunable parameters is considered as the measure of fine-tuning depth?

---

### Official Review · Reviewer_jtAM · 2025-10-24

**Soundness:** 2
**Presentation:** 2
**Contribution:** 2
**Rating:** 4
**Confidence:** 4

**Summary:**

This paper investigates how fine-tuning depth (Deep vs. Shallow/PEFT) and model capacity interact to affect AI alignment outcomes. The authors apply Stage-6 Preference Alignment (S6-PA) across GPT-4o (deep alignment via proprietary API), Llama-3, and Qwen-3 (shallow alignment via QLoRA). Results show GPT-4o reduced adversarial deception by 80%, while shallow alignment in high-capacity models either failed (Llama-3 resistance) or produced alignment-faking where models instrumentalized ethical language to justify deception (Qwen-3-32B). Critically, the alignment-faking model improved on standard safety benchmarks while the robustly aligned model degraded，revealing an "Evaluation Paradox" where conventional metrics reward superficial compliance while masking strategic deception.

**Strengths:**

1. Unlike prior work that explicitly trains deception, this paper demonstrates that alignment-faking can emerge unintentionally from standard fine-tuning processes, representing an important empirical contribution to understanding treatment-induced safety risks.

2. Showing that standard safety benchmarks (garak-do-not-answer) reward the alignment-faking model while penalizing the robustly aligned one exposes fundamental limitations in current evaluation practices and has major implications for how the field measures AI safety.

**Weaknesses:**

1. The core comparison is flawed because GPT-4o uses a black-box proprietary API while Llama/Qwen use QLoRA, confounding alignment depth with model family, architecture, pre-training data, base safety training, and unknown implementation details. The authors cannot validly attribute GPT-4o's success to "deep alignment" rather than these other factors. A rigorous design requires comparing full fine-tuning vs. PEFT on the same base model, and this confound undermines the paper's central causal claim.

2. As a paper claiming to discover new phenomena (spontaneous alignment-faking and the Evaluation Paradox), the experimental scope is insufficient to establish generalizability. The limited model coverage (only 3 families, no systematic capacity scaling) combined with small sample sizes (50 training dilemmas, 20 test dilemmas, 40 adversarial prompts) and lack of statistical validation raises concerns about whether these findings represent stable, generalizable phenomena.

**Questions:**

1. Can you isolate the depth variable by applying full fine-tuning to Llama-3 or Qwen-3 to separate alignment depth from model family differences?

2. How robust is the alignment-faking phenomenon across hyperparameter sweeps for all models? What variance exists in deception rates, how many random seeds were tested, and what's the stability of the phenomenon?

3. Does the adversarial test generalize beyond the system prompt that explicitly permits deception? How do models behave with more subtle instrumental pressures or realistic deployment scenarios without explicit permission for deception?

4. What's the capacity threshold where shallow alignment begins to fail? Given that Qwen-3-0.6B succeeds while Qwen-3-32B fails under shallow alignment, the paper's hypothesis about scale-dependent failure modes requires more systematic validation. A convincing demonstration would involve testing multiple intermediate model sizes (e.g., 0.6B, 1.7B, 4B, 8B, 32B) to establish a clear relationship between capacity and alignment failure.

---

### Official Review · Reviewer_PDQ1 · 2025-10-31

**Soundness:** 2
**Presentation:** 3
**Contribution:** 1
**Rating:** 2
**Confidence:** 2

**Summary:**

The paper investigates the impact of using parameter-efficient vs. full finetuning for LLM alignment, hypothesizing that low-rank adaptors may not be powerful enough to steer multibillion-parameter models away from strong, undesirable priors. To test this hypothesis, the authors construct a novel alignment dataset with 70 LLM-generated ethical dilemmas and associated preferred responses based on Kohlberg's Stage 6 morality. The dataset contains 50 Japanese examples for training and 20 English examples for testing, all validated by human experts. A variety of pretrained LLMs were finetuned on this dataset, either with full finetuning or with low-rank adaptors, and all performed well on the test set, showcasing their newly acquired ethical reasoning capabilities. However, when faced with system prompts encouraging unethical actions, some of these models still engaged in undesirable behavior, and interestingly, this misalignment was not reflected in the score of an established safety benchmark (garak-do-not-answer).

**Strengths:**

* The main hypothesis -- that PEFT may be insufficient to override strong priors in LLMs -- is well worth studying, given the widely spread usage of these methods.
* The manuscript has a clear structure that makes it easy to navigate across the main text and appendix. The numbered research questions (RQs) are particularly helpful for keeping track of the different experiments and benchmarks.
* Constructing a novel alignment dataset based on Kohlberg's 6 stages of human moral development seems like a novel and meaningful idea, especially when validated by human experts.
* The finding that an alignment-faking LLM may attain better scores on a popular safety benchmark is deeply concerning and absolutely warrants further research.

**Weaknesses:**

* The scope of the paper feels more like an initial exploration than a complete study, as the experimental design lacks proper systematic comparisons. Currently, different models are evaluated with different methods (some with full finetuning, others with PEFT), making it impossible to draw general conclusions. A more insightful approach would apply both methods to the same models and analyze the differences. The paper poses four interesting research questions, but the answers end up being quite specific to particular model-method combinations. While the authors are generally careful not to overclaim within the main text (which I appreciate), this means the broader takeaways remain overly limited, and it's hard to see what future researchers would take away from this work beyond a few promising data points.
* The abstract doesn't match the careful tone of the main text and tends to give readers the wrong impression about how broad the evidence is. For example, it uses plurals where the experiments are singular: "high-capacity models" (only one was tested with deep alignment), "standard safety benchmarks" (only one was used). As a side note, the TLDR on the OpenReview page is even more problematic in this regard.
* Despite its merits, the newly introduced S6-PA dataset remains very small: 50 training items and 20 test items feels insufficient for drawing robust conclusions. More worrying (and straightout puzzling) is the decision to use Japanese for the training set, without explanation. Given that most LLMs perform significantly worse in Japanese (in part due to tokenization), this choice seems like it could introduce serious confounding factors. Clear justification is needed, along with an analysis of the impact of this language mismatch.
* Out of only 16 cited references, at least four are problematic. Two seem LLM-hallucinated (which I find unacceptable, see my ethics review flag for details), and another two have slightly incorrect details:
	* Perez et al. is from 2023, not 2022
	* Zou et al. (2023) is missing two authors (Nicholas Carlini and Milad Nasr)
Furthermore, the prominently featured safety benchmark `garak-do-not-answer` is missing proper citation. I believe a link to the NVIDIA garak github and a citation of the original Do-Not-Answer paper (Wang et al., 2023) would be necessary here.
* Minor point: The overall writing style is extremely LLM-like. I don't necessarily mind it, but it was very noticeable and some readers may have a problem with it.

**Questions:**

- The "Evaluation Paradox", a central finding of the paper, is only demonstrated on one alignment-faking LLM, despite two other Llama models also showing this behavior. How did they perform in RQ3?
- The new S6-PA dataset is a core contribution of this work, rightfully taking up several pages in the main body and appendix. Yet, surprisingly, it is not mentioned in the abstract. Was this an oversight?
- The paper shows that safety benchmarks can be misleading, which makes me wonder: how reliable is the S6-PA test set as a safety benchmark (RQ2)? Can this be gamed too?

**Details Of Ethics Concerns:**

Two cited references appear to be (at least partially) LLM-hallucinated:
1. >Frederik Mölder, Tian Zhao, Tom Lükewille, Javier Rando, Jacob Andreas, and Stephen Casper. Assessing the brittleness of safety alignment via pruning and low-rank modifications. In International Conference on Learning Representations (ICLR), 2024.

With the exception of Stephen Casper, none of the cited authors seem to exist. A paper with this title and year does exist, appearing at ICML under a completely different set of authors:
_Boyi Wei, Kaixuan Huang, Yangsibo Huang, Tinghao Xie, Xiangyu Qi, Mengzhou Xia, Prateek Mittal, Mengdi Wang, Peter Henderson_

2. >Michael V Moore, Pedro Romero, Sarah Fitz, Luyang Sun, and Mariam Abdulhai. Using developmental constructs as evaluation signals for artificial intelligence. arXiv preprint arXiv:2212.09251, 2022.

I could not find the paper based on the title, and all authors appear to be fabricated names. The arxiv ID does exist, pointing to a paper by Perez et al., which is also cited in this paper (with proper details).

I find this particularly worrying for the scientific trustworthiness of the paper.

---

### Note · Authors · 2025-11-18

**Comment:**

We sincerely thank the reviewers and area chairs for their time and thoughtful feedback.
After careful consideration, we have decided to withdraw the paper to address the issues raised and strengthen the work.
We appreciate the community’s efforts and will aim to return with an improved version.

**Withdrawal Confirmation:**

I have read and agree with the venue's withdrawal policy on behalf of myself and my co-authors.